# Exploring Peptaibol’s Profile, Antifungal, and Antitumor Activity of Emericellipsin A of *Emericellopsis* Species from Soda and Saline Soils

**DOI:** 10.3390/molecules27051736

**Published:** 2022-03-07

**Authors:** Anastasia E. Kuvarina, Irina A. Gavryushina, Maxim A. Sykonnikov, Tatiana A. Efimenko, Natalia N. Markelova, Elena N. Bilanenko, Sofiya A. Bondarenko, Lyudmila Y. Kokaeva, Alla V. Timofeeva, Marina V. Serebryakova, Anna S. Barashkova, Eugene A. Rogozhin, Marina L. Georgieva, Vera S. Sadykova

**Affiliations:** 1Laboratory for Taxonomic Study and Collection of Cultures of Microorganisms, Gause Institute of New Antibiotics, St. Bolshaya Pirogovskaya, 11, 119021 Moscow, Russia; nastena.lysenko@mail.ru (A.E.K.); irina-alekcandrovna2013@yandex.ru (I.A.G.); sukonnikoff.maxim@yandex.ru (M.A.S.); efimen@inbox.ru (T.A.E.); nataljamarkelova@yandex.ru (N.N.M.); rea21@list.ru (E.A.R.); 2Faculty of Biology, Lomonosov Moscow State University, 1-12 Leninskie Gory, 119234 Moscow, Russia; e_bilanenko@mail.ru (E.N.B.); bond.sonia@gmail.com (S.A.B.); kokaeval@gmail.com (L.Y.K.); 3Belozersky Institute of Physico-Chemical Biology, Lomonosov Moscow State University, 119991 Moscow, Russia; 2.04.cochon@gmail.com (A.V.T.); mserebr@mail.ru (M.V.S.); 4Shemyakin and Ovchinnikov Institute of Bioorganic Chemistry, RAS, St. Miklukho-Maklaya, 16/10, 117997 Moscow, Russia; barashkova.an@gmail.com

**Keywords:** alkaliphilic fungi, peptide, antifungal activity, antitumor activity, *Emericellopsis alkalina*, emericellipsin’s complex

## Abstract

Features of the biochemical adaptations of alkaliphilic fungi to exist in extreme environments could promote the production of active antibiotic compounds with the potential to control microorganisms, causing infections associated with health care. Thirty-eight alkaliphilic and alkalitolerant *Emericellopsis* strains (*E. alkalina*, *E.* cf. *maritima*, *E.* cf. *terricola*, *Emericellopsis* sp.) isolated from different saline soda soils and belonging to marine, terrestrial, and soda soil ecological clades were investigated for emericellipsin A (EmiA) biosynthesis, an antifungal peptaibol previously described for *Emericellopsis* *alkalina*. The analysis of the *Emericellopsis* sp. strains belonging to marine and terrestrial clades from chloride soils revealed another novel form with a mass of 1032.7 Da, defined by MALDI-TOF Ms/Ms spectrometers, as the EmiA lacked a hydroxyl (dEmiA). EmiA displayed strong inhibitory effects on cell proliferation and viability of HCT 116 cells in a dose- and time-dependent manners and induced apoptosis.

## 1. Introduction

Fungi of the *Emericellopsis* genus are distributed worldwide in terrestrial and marine environments. However, it is in saline environments that *Emericellopsis* species appear more frequently on substrates such as macroalgae and sponges, in sea and estuarine waters, marine sediments, and are often in soils with periodic flooding, extreme humidity, and alkalinity [1,2,3,4,5]. Currently, there are 27 entries listed in the genus *Emericellopsis* (Hypocreales, Sordariomycetes, Ascomycota) in the Index Fungorum including 23 species and four varieties [6]. The study of alkaliphilic isolates transformed the existing ideas concerning two distinct ecological clades within *Emericellopsis*, one consisting of terrestrial isolates and the other predominantly marine. Alkaliphilic *Emericellopsis alkalina* from soda soils seem to have been derived from a marine lineage and are nested in their own subclade within the marine clade [1,3,4,5]. According to recent research of multi-locus gene phylogenies, these species are placed in three ecological clades: terrestrial, marine, and soda soil.

From chemical studies, it is known that species within the genus *Emericellopsis* can produce a range of known bioactive metabolites to possess antimicrobial activity against plant and human pathogens and anticancer activity [1,3,4,5,7,8,9,10].

The Norine database [11], which is dedicated to non-ribosomal peptide synthases (NRPS), includes five peptaibol complexes from *Emericellopsis* species. Untargeted metabolomics using UPLC-QToF–MS/MS and genome sequencing (Illumina HiSeq) has been performed for the recently described new species *E. cladophorae* and *E. atlantica*. Their genomes contain genes involved in the biosynthesis of a range of bioactive metabolites [3,12,13]. Despite evidence of secondary metabolite production, our understanding of the full biosynthetic potential of *Emericellopsis* species remains limited. 

In this study, peptaibol profiles were analyzed to investigate the role of peptaibols in the antagonistic effect of *Emericellopsis* species from different clades. In a continuation of our investigations into chemodiversity and the yield of antimicrobial peptaibols, we focused this study on the peptaibol emericellipsin A production of the *E. alkalina*, *E.* cf. *maritima*, *E.* cf. *terricola*, and *Emericellopsis* sp. derived from soda and saline environments.

## 2. Results

*Emericellopsis* fungi are among the most important dominants on the coast of soda and salt lakes. In general, soils with a soda type of salinity are characterized by a lower content of fungal propagules, with the dominance of alkaliphilic isolates of fungi *Emericellopsis alkalina* and a minimum number of other *Emericelopsis* taxa. These fungi are also found on the edge of lakes with other types of salinity in which the amount of soluble salts remains high, but the pH decreases significantly. All 38 strains were isolated from samples of soils collected on the edge of numerous soda and saline lakes of the Kulunda Steppe (Altai, Russia) over several years. Twenty-two isolates were previously named *E. alkalina* and showed high phylogenetic similarity based on six loci (LSU and SSU rDNA, RPB2, TEF1-α, β-tub, and ITS region) [1]. Additionally, we studied 16 *Emericellopsis* strains isolated from soils in 2017–2018. Fungal identification was carried out using a polyphasic approach (morphological and molecular). All our new strains showed asexual acremonium-like sporulation.

Thus, the sequences generated in this study and sequences from *Emericellopsis* species deposited in GenBank were included in the phylogenetic analyses. Phylogenetic analyses were conducted using ITS rDNA region sequences of 62 strains of fungi included with sequences of ex-type cultures of *Emericellopsis* species from GenBank (Appendix A). Data for 14 *E. alkalina* strains from the collection are not shown in the final tree as they were previously completed [1]. Our new *Emericellopsis* strains were grouped in three ecological clades: 10 strains *E. alkalina* in “soda soil” clade; four strains (1KS17-1, 3KS17-1, 3KS17-2, and 3KS17-12) were in the “marine” clade; and two strains (1KS17-4, 2KS17-1) in the “terrestrial” clade (Figure 1). Two strains (1KS17-1 and 3KS17-1) in the marine clade were identified as *Emericellopsis* cf. *maritima*, as they grouped within the known ex-type strain of *E. maritima* (CBS 491.71). Strain 1KS17-4 in the terrestrial clade was identified as *Emericellopsis* cf. *terricola*, as it grouped within the known ex-type strain of *E. terricola* (CBS 124.42). The remaining new strains from saline soils formed their own clades in the *Emericellopsis* lineage, not falling into known species. We accept these as *Emericellopsis* sp.

A link between culture growth rate and pH has previously been observed and showed that the pH preferences varied among the members of the different clades within the *Emericellopsis* genus [1,3]. All 32 *E. alkalina* strains described here showed good growth ability from neutral with pH 6 to an alkaline medium with pH 10, had an optimum growth at pH 9 and displayed an alkaliphilic phenotype. Other strains showed an alkalitolerant phenotype. Four members of the marine clade had an optimum growth at pH 7 (3KS17-1, 3KS17-2, and 3KS17-12) or pH 8 (1KS17-1), but reduced growth at higher pH values. Two members (1KS17-1 and 2KS17-1) of the terrestrial clade had different optimum pH values. The strain *Emericellopsis* sp. 2KS17-1 had an optimum growth at pH 8 and was able to tolerate higher pH values. The strain *Emericellopsis* cf. *terricola* 1KS17-4 had an optimum growth at pH 7 and reduced growth at higher pH values (Appendix A).

### 2.1. Biosynthesis of Emericellipsin A and Its Isoform by Different Emericellopsis Species

In the study of the effect of three liquid media on the production of peptaibols from 38 *Emericellopsis* sp., it was observed that the production of all of the peptaibols were not the same on different media. It was found that emericellipsins failed to appear in PDB and Chapek–Doks media in all of the investigated strains. We estimated the production of bioactive EmiA for all *E. alkalina* to increase dramatically only on alkaline media in the culture broth as well as in mycelium (Table 1, Appendix A). The results of the antifungal activity of the crude emericellipsin’s complex extracts obtained with the *E. alkalina* and two *Emericellopsis* sp. strains tested showed that it inhibited the growth of fungal tests in the agar diffusion assay, as shown in Appendix A. 

Comparative analyses of major compound EmiA detected significant differences between the strains from different saline environments: for 20 of the isolates, the amount of target peptaibol predominated in the mycelium. A high content of EmiA (more than 200 mg/L) was found in nine isolates in the mycelium and in five isolates in the culture broth. The *E. alkalina* A113 isolate from soda soils and *E. alkalina* M20 from soda-chloride soils were characterized by high levels of EmiA both in the culture broth and in the mycelium, amounting to 358.75 and 356.5 mg/L for A113 and 342, respectively, and 202 mg/L for M20. The maximum amount of EmiA was found in the culture broth of the ex-type strains of *E. alkalina* E101—429.5 mg/L. 

The isolates of *E. alkalina* from soda soils were characterized by the high accumulation of EmiA as well as isolates from soda chloride-sulfate, and chloride-sulfate soils. The maximum values for EmiA were noted in isolates of *E. alkalina* from soda soils. Other *Emericellopsis* species were isolated only from chloride soils. Two strains (1KS17-1 and 2KS17-1) produced EmiA when cultivated on the alkaline medium, and the amount of EmiA was higher in mycelium than in the culture broth. There were no detected antifungal activities for other *Emericellopsis* sp. and no emericellipsin production was detected. The result of a comparative study of emericellipsin production indicated that most of the isolates with a good yield of EmiA also produced emericellipsins B–D in the mycelium. None of the isolates derived from chloride and chloride-sulfate soils produced the homologous B–D.

The analysis of the two *Emericellopsis* sp. strains (1KS17-1, 2KS17-1) belonging to the marine and terrestrial clades from chloride soils revealed similar, but still different, profiles of emericellipsins. Further purification of its crude extract resulted in the detection of other novel forms of emericellipsin complexes, not detected as emericellipsins B–E by the HPLC and MALDI-TOF MS/MS methods. The antimicrobial assay showed that the compound had strong inhibitory effects on *Aspergillus niger* and *Candida albicans* (Appendix A). Results revealed from the bioautographic visualization showed two growth inhibition zones with Rf means: 0.2 and 0.55 correspondingly in the system of CHCl3:MeOH = 3:1, and there was a zone of growth inhibition. Most of the fragment’s molecular differences were close to correlating with each other with a little mass difference at ±1 Da (probably the same), except for one monomeric fragment with a mass 195 Da in the fragmentation spectra of molecular ion with a mass of 1032.7 Da, and 213 Da in the fragmentation spectra of molecular ions with a mass of 1050.7 Da (Figure 3a).

### 2.2. Isolation and Characterization of Emericellispsin A and DE Hydroxyl Form

Analytical reverse-phase HPLC was used to clarify the degree of molecular diversity of fungal metabolites that are chemically close to peptaibols. Compounds that are hydrophobic to different degrees (based on the retention time in separation by liquid chromatography) were found in the elution zone of the main component, emericellipsin A. Previously, this was shown among the entire spectrum of peptaibol compounds in addition to B–E compounds [14]. The fraction was also analyzed with the HPLC method. HPLC analysis results were obtained and only two peaks 1 (t_R_ = 37.3 min) and 2 (t_R_ = 39.4 min) were visualized in the chromatogram (Figure 2a). 

These two peaks were separately collected and analyzed with the MALDI-TOF MS method. The major peak with a molecular ion mass of 1050.7 Da was detected to be related to chromatographic peak 1 (Figure 2b); 1032.7 Da and 1050.7 Da were the major and the minor characteristic molecular ion masses, respectively, correspondingly related to substances included in chromatographic peak 2 (Figure 2c). All masses were defined as positively charged [M + H]^+^ ion masses. Based on the MALDI-TOF MS results, it can be concluded that ions with masses of 1032.7 Da and 1050.7 Da related to the main components of the analyzed sample. The molecular ion mass of 1050.7 Da is more likely to be related to EmiA, because it was the same with the computational mass meaning caused by its structure (Figure 3). 

The substance related to the ion mass of 1032.7 Da was undefined with any information absent at the Emericellopsis alkalina metabolite bibliography with the same mass. To obtain more information on both substances, they were analyzed in the fragmentation assay with the MALDI-TOF MS/MS method. First, it should be noted that the difference between two molecular ion masses in 18 Da correlated with the molecular mass of water molecules (Mr~18 Da) (Figure 2b,c).

In addition, the same character of the fragment’s mass distribution indicated the same nature of both compounds (Figure 3a). All of the fragment ion masses were compared with each other, and the molecular differences between fragment ions related to the possible residues in molecular structure of EmiA were revealed.

Moreover, the chemical structure of the AHMOA residue involves a secondary hydroxyl situated through one carbon atom to the amide bond, and so this is probably a good site for water elimination. If a hydroxyl is lacking, it must form a double bond between the corresponding carbon atoms for the valency principle to be conserved. Thus, the monomeric fragment with a mass of 195 Da is more likely to relate to the 2-amino-4-methyl-8-oxodec-6-enoic acid residue (Figure 3b). To sum it up, the presence of EmiA related to the molecular ion with a mass of 1050.7 Da was confirmed with fragmentation spectra, as was the nature of molecular ion with a mass of 1032.7.

Thus, in the analyzed sample, we observed two components with retention times of 37.3 min and 39.4 min in the chromatogram, correspondingly. Both peaks 1 and 2 were related to EmiA and dEmiA, respectively. dEmiA was first observed and was characterized as a more hydrophobic analogue of EmiA lacking a hydroxyl.

Bioactivity-based assays of the dEmiA form detected activity on Aspergillus niger ATCC 16404 and Candida albicans ATCC 14053 fungi with MIC 4 and 2 µg/mL. The same results have been previously described for EmiA [8,14]. Since both forms of EmiA and dEmiA provide bioactivity, it can be considered that the single secondary hydroxyl-group of EmiA does not play a crucial role in the mechanism of antibiotic action. Collecting information on dEmiA, it can be considered that dEmiA is the premature form of EmiA.

### 2.3. Effects of Emericellipsin A Peptide on Colorectal Carcinoma (HCT116) Cells in Real-Time Systems 

The previous results showed that EmiA also has cytotoxic properties toward a wide range of tumor cell lines of various origins. A distinctive feature of EmiA is the contrast pattern of cell growth inhibition, which is expressed in a significant variation in the IC_50_ index. The maximum activity was demonstrated with respect to the cancer HCT116 line (1 μM) [14]. To determine the effect of EmiA on the growth of HCT116 cells, we used the iCELLigence real-time cell analysis system (RTCA) so that we could not only monitor the growth inhibitory effect of peptaibol, but also its effects on other cellular changes. This is useful in the selection of further methods to estimate the compound action and helps us to better understand the mechanisms of toxicity and supports the selection of the best compound candidates in early drug development before entering animal testing.

The CI represents a quantitative measure of the proliferation, cell adhesion, and overall viability of the assessed cells. After 24 h, EmiA was applied. The viability, adhesion, and proliferation of the cells were monitored before and during treatment in real-time for 72 h (Figure 4a). During the first day, CI increased, which corresponded to the process of cell adhesion and proliferation in culture. After updating the medium in wells and introducing various concentrations of EmiA, CI began to decrease compared to the control. Differences in cell proliferation under the influence of increasing concentrations of EmiA caused changes in CI. After the introduction of the peptide and during the subsequent period of incubation, CI changed according to the concentrations of the peptide: in the control, from 0.4 to 1.3; at 0.25 μg/mL—from 0.4 to 1.1; at 1.0 μg/mL—from 0.4 to 0.7; at 4.0 μg/mL—from 0.4 to 0.6; at 16.0 μg/mL—from 0.4 to 0.5. EmiA exhibited a strong inhibitory effect on HCT116 cell lines and retained their CI values continuously at low levels at concentrations ≥1.0 μg/mL.

Cytotoxic agents, which inhibit cell growth and division by suppressing proliferation, causing cell cycle arrest in the G0/1 and G2/M phases, are used in the treatment of many types of cancer; in addition, cell growth is suppressed through the induction of apoptosis [15]. Screening of substances and their concentrations in the RTCA system in relation to tumor cells makes it possible to determine the lowest concentration of effectors necessary for further study of their cytotoxic effect including morphological and functional changes in cells as well as the degree of cell death. Subsequent analysis of HCT116 cells by flow cytofluorometry after 24 h of exposure to EmiA at a concentration of 0.25 μg/μL revealed apoptosis and led to an accumulation of DNA content in the G0/G1 phase (Figure 4b). The results of the real-time impedance method were comparable to those of the cytotoxicity assays in many studies [15,16]. In our study of the RTCA system, preliminary screening of the antitumor activity of the EmiA revealed a peptide concentration of 1.0 μg/mL, which effectively reduced CI, and a peptide concentration of 0.25 μg/mL, which minimally inhibits colorectal carcinoma (HCT116) cells.

## 3. Discussion

Invasive aspergillosis (IA) is a well-known complication of intensive cytostatic and immunosuppressive therapy, organ, and tissue transplantation. However, at present, the risk group for developing IA has significantly expanded to encompass other categories of patients including patients in intensive care units (ICUs) with severe influenza and COVID-19. Opportunistic fungal pathogens have emerged as a leading cause of human mortality with attributable mortalities estimated at ~1.5 million per year [17,18]. 

Antimicrobial peptides (AMPs) have aroused great interest as potential next-generation antifungal drugs since they are bioactive small proteins, naturally produced by all living organisms, and represent the first line of defense against fungi [10,17,19,20,21]. 

Peptaibols, the largest group of peptaibiotics, are a class of linear peptides that have an acylated N terminus group, a C-terminal amino acid, and a high content of α-aminoisobutyric acid (Aib)—approximately 40% of Aib in long peptaibols and from 14 to 56% in short peptaibols [7,11,22,23,24,25,26,27,28]. Peptaibols are dominant secondary metabolites of filamentous fungi including the ones exhibiting a mycoparasitic lifestyle [22,24,25,29,30]. Many authors have described peptaibols with antifungal activity, but most of them possessed activity against plant pathogens and fungi and are currently receiving significant attention because of their applications as bio fungicides [19,25]. In recent years, a few studies have obtained the activity of peptaibols against clinical pathogenic fungi and cancer lines [5,11,14,25,31,32,33,34].

Today, about 30 genera of filamentous fungi, mostly belonging to the order of Hypocreales, have been recognized as promising sources of peptaibol groups [13,14,19,21,34,35,36]. A majority of them have been discovered in the fungal species of the genus *Trichoderma*. The *T. viride* clade, *T. brevicompactum* clade, *T. virens*, *T. parceramosum/T. ghanense*, and *T. longibrachiatum* clades have been the most intensively studied species for the synthesis of peptaibols [29,37,38,39]. The fungi of the *Emericellopsis* also produced antimicrobial peptides of the peptaibol group: zervamicins, bergofungins, emerimicins, and emericellipsins have been reported for eight species of these fungi [7,11,13]. 

The genus *Emericellopsis* harbors 23 species described thus far. Recently, Hagestad et al. [3,5,13,14,40] sequenced the first genome of an *Emericellopsis* species: *E. atlantica* strain TS7. The genome assembly and gene statistics were carried out for *E. cladophorae* and *E. atlantica*. Both genomes shared several conserved genes in biosynthetic gene clusters of ascochlorin, leucinostatin A/B, and cephalosporin C. However, some biosynthetic gene clusters (BGCs) were species specific such as those encoding for helvolic acid, botrydial, and fusaristatin A in *E. atlantica*, while clavaric acid, EQ-4, squalestatin S1, and (-)-Mellein in *E. cladophorae*. The fungi of the genus *Emericellopsis* are well known for their production of peptaibols with antibacterial and antifungal activity: the antiamoebins I–XI from *E. salmosynnemata* and *E. synnematicola*, the bergofungins A–D from *E. donezkii*, the emerimicins II, III, IV from *E. microspora* and *E. minima*, the heptaibin from *Emericellopsis* sp., the zervamicines from *E. salmosynnemata*—32 peptides grouped in five families according to the Norine databases [41]. 

The discovery of novel peptaibols from marine fungi, fungi habitats from saline and soda soils, and other unique locations offer further antibiotic discovery findings from natural sources [3,10,12,21,34,36,40,42]. Our previous work on peptaibol chemodiversity produced by *E. alkalina* from soda soils led to the identification of new peptaibol complexes named emericellipsin A–E with broad antimicrobial activities [1,4,8,9,14]. The main compound EmiA demonstrated strong antifungal activity against clinical isolates of *Aspergillus terreus* 1133 m, *A. fumigatus* 163 m, *A. ochraceus* 497 m, *Saccharomyces cerevisiae* 77 m, and *Cr. laurentii* 801 m, with multi-drug resistance toward fluconazole and amphotericin B [14]. 

The influence of media components as well as pH is well-known as a key factor in the synthesis of antimicrobial metabolites by producers from different taxa [22,43]. Medium composition can act as a multiplied aspect toward the good or poor growth of fungi by supplying adequate local intracellular concentration of the direct precursor amino acids. The *Trichoderma* spp. strain RK10-F026 appeared to be culture condition-specific, where culture compositions activate groups of peptaibols [44]. During investigations on peptaibol chemodiversity from marine-derived *Trichoderma* spp., five new 15-residue peptaibols named pentadecaibins I–V (1–5) were isolated from the specific solid culture of the strain *Trichoderma* sp. MMS1255 belonging to the *T. harzianum* species complex [45]. In the present study, by analyzing the effect of the crude peptaibol extract, it was found that emericellipsins failed to appear in PDB (potato dextrose broth) and Chapek–Doks media in all investigated *Emericellopsis* strains. We estimated the production of the major compound EmiA for all *E. alkalina* on alkaline media to increase dramatically in the culture broth as well as in the mycelium.

Optimization is an important point for the production of desirable metabolites from microbial strains [14,43]. Lower production of the desired metabolites has always been a bottleneck in carrying out purification steps for further studies. It is noteworthy in this study that different amounts of emericellipsins were produced in different culture conditions. Two isolates from *Emericellopsis* sp. strains from chloride soils also produced EmiA when cultivated on an alkaline medium. Regarding the results of two other taxa in *Emericellopsis* sp., there were no detected antifungal activities and no peptaibol production was found.

Our strategy to analyze other *Emericellopsis* sp. from different soda and saline biotopes with varying culture compositions proved to be useful in identifying new compounds. The analysis of the *Emericellopsis* sp. strains (1KS17-1, 2KS17-1) belonging to marine and terrestrial clades from chloride soils revealed another novel form with a mass of 1032.7 Da\as defined by MALDI-TOF Ms/Ms spectra as the EmiA lacking a hydroxyl (dEmiA). Bioactivity-based assays of this dEmiA form detected activity on *Aspergillus niger* and *Candida albicans* fungi with MIC 4 and 2, respectively. Our previous results showed the same MIC (4 and 2 µg/mL) for the main EmiA in these tests [8,14]. As different culture compositions can contribute to the distribution of peptaibol groups, it is in our best interest to further mine their biosynthetic processes in the future. 

Some peptaibols have also been proposed as antitumor molecules [31,32,33,46]. Indeed, they display selective cytotoxicity toward cancer cells, which has been ascribed to the different composition and physical properties of the cancer cell membrane relative to healthy cells [20,25,31,33,46,47]. Reinvestigation of the well-known peptaibol producer *Trichoderma arundinaceum* strain MSX70741 to discover new anticancer compounds led to the isolation of three new peptaibols with antitumor activity. The cytotoxic activities of the new compounds were evaluated against a panel of human cancer cell lines: HCT 116, DLD-1, HT-29, and SW948, Hep-G2, and Huh-7, and HeLa. Trichobrevin BIII-D exhibited moderate activity against cell HCT 116 and HT-29 with IC_50_ values of 6.8 and 6.7 mM, respectively [24,25,29].

In our previous results, EmiA displayed strong inhibitory effects against cell lines HCT 116 and Hela [14]. Therefore, it is less toxic to normal cells than doxorubicin (~40 times), but yields a more potent cytotoxic effect on tumor cell lines. In the present analysis, EmiA decreased the proliferation and viability of the HCT116 colorectal carcinoma cell line in vitro in the iCELLigence RTCA system [15,16,48,49]. The most pronounced activity of the peptide was found at concentrations ≥ 1.0 μg/mL. Notably, EmiA decreased the CI of HCT116 colorectal carcinoma cell for more than three days in real-time monitoring. In summary, this in vitro study demonstrated that EmiA could inhibit the proliferation and viability of different cancer cell lines, and as detected for the HCT116 line, suppressed cycle arrest in the G0/1 and G2/M phases. EmiA displayed strong inhibitory effects on the cell proliferation and viability of HCT 116 cells in dose- and time-dependent manners and induced apoptosis.

Our results, together with previous information regarding the effect on pathogenic fungi and cancer cells, show that lipopeptaibols EmiA from the alkaliphilic fungus *E.* *alkalina* is a promising treatment alternative to licensed antifungal drugs for invasive mycosis therapy for multidrug-resistant aspergillosis and cryptococcosis. EmiA was similar to that of amphotericin B against drug-resistant pathogenic fungi. Finally, EmiA is not meant to replace doxorubicin-based chemotherapy in patients but could be a potential mild therapeutic option for patients with aspergillosis who are not suitable for chemotherapy and have to undergo palliative treatment.

## 4. Materials and Methods

### 4.1. Fungal Strain Identification and Phylogenetic Analysis

All 38 strains of fungi were isolated from soils on the edge of soda or saline lakes of the Kulunda Steppe (Altai, Russia), as previously described [1]. Among them, 22 *Emericellopsis alkalina* strains were received from the Collection of Fungi from the Extremophilic Habitat, Lomonosov Moscow State University. Species identification was conducted by molecular-genetics methods based on the sequence data of ITS rDNA, LSU rDNA, SSU rDNA, TEF-1α, β-tub, and RPB2 [1]. Some of the isolates were deposited at the CBS-KNAW Fungal Biodiversity Center (Utrecht, The Netherlands), the All-Russian Collection of Microorganisms (VKM, Pushchino, Russia), and the Russian National Collection of Industrial Microorganisms (VKPM). Additionally, we studied 16 *Emericellopsis* strains isolated from soils in 2017–2018. Fungal morphological traits (such as septate hyphae, conidiophores, conidia, and another) were examined under a light microscope and a scanning electron microscope (JEOL JSM-6380LA). Molecular identification of fungal strains was performed based on PCR amplification of the internal transcribed spacer rDNA (ITS) region of fungal rDNA using primers ITS1f and ITS4r. The genomic DNA was isolated using a set of DNeasyPowerSoil Kit reagents (Qiagen Inc., Carlsbad, CA, USA) following the manufacturer’s instructions. The final volume of the 50 µL PCR mix included 25 µL of 2X PCR Master Mix (ThermoScientific, Waltham, MA, USA) 0.5 µM of each primer, 1–100 ng of isolated DNA, and water (nuclease-free). PCR was performed according to the following scheme: (1) 94 °C—5 min; (2) 33 cycles with alternating temperature intervals 94 °C—1 min, 51 °C—1 min, 72 °C—1 min; and (3) 72 °C—7 min. The DNA fragments were sequenced by the Sanger method on the Applied Biosystems 3500 Series Genetic Analyzer. Newly generated sequences have been deposited in GenBank [50] with corresponding accession numbers (Appendix A). To analyze the taxonomy of our strains, we performed phylogenetic reconstruction including 24 sequences *Emericellopsis* from ex-type strains or cultures (Appendix A). Alignments were calculated through the MAFFT v. 7.429 online server [51] using the L-INS-I strategy. Alignments were carefully checked visually and manually modified; all ambiguous sections were excluded from the analysis. Phylogenetic trees were inferred using the maximum likelihood (ML) method. Before the analyses, the best-fit substitution model for the alignment was estimated based on the Akaike information criterion (AIC) using the IQ-TREE Web Service [52]. The TIM2e plus Gamma model was chosen for the ITS dataset. The RAxML program ver. 7.0.3 was used for the heuristic search. 

### 4.2. Cultivation of the Fungi and Extraction of Emericellipsins 

Different growth media were used for the optimal production of peptaibols. The media were (i) potato dextrose broth (PDB); (ii) alkaline media based on phosphate buffers (pH 10 (±0.2); and (iii) Chapek–Doks media according to the previous protocol [4,14,53]. The effect of pH on the growth rate was evaluated in triplicate on plates with several growth media based on citrate, phosphate, and carbonate buffers (pH 4, 5, 6, 7, 8, 9, 10 (±0.2)). The contents of the alkaline media and buffers composition were followed by Grum–Grzhimaylo et al. [1]. For the production of EmiA, homologous fungi were cultivated at 26 °C in Erlenmeyer flasks under stationary conditions for 14 days [8]. The culture fluid (CF) was separated by filtration through membrane filters on a Seitz funnel under a vacuum. To isolate the antibiotic substances, the CF of the producers was extracted three times with ethyl acetate in an organic solvent/CF ratio of 1:5. The obtained extracts were evaporated in a vacuum on a Rotavapor rotary evaporator (Buchi, Flawil, Switzerland) to dryness at 42 °C, the residue was dissolved in aqueous 70% ethanol, and the alcohol concentrates were obtained [14]. 

### 4.3. Purification and Identification of Emericellipsin A and Dehydrate Isoform 

#### 4.3.1. HPLC analysis of Prepared Samples

Isolation of the peptaibol from the extract was achieved by HPLC. Chromatographic (HPLC) analysis was performed using a Milichrom A-02 (ZAO Econova, Novosibirsk) chromatograph, Nucleosil-100-5-C18 column (Macherey-Nagel, Düren, Germany; L = 75.0 mm; D = 2.0 mm; d = 5 μm). Wavelength of detection used was 214 nm. Mobile phase included: A—H_2_O (MQ) + 0.02% TFA (HPLC, Sigma-Aldrich, Darmstadt, Germany); B—MeCN (HPLC, Fisher Chemicals) +0.02% TFA (HPLC, Sigma-Aldrich, Germany). Injection volume was 15 μL; flow rate—100 μL/min; T = 35 °C. Gradient program X was used in workflow: X—from 0 to 100% B for 46 min.

#### 4.3.2. Calibration and Quantitation of Emericellipsin A

This procedure was carried out using analytical reversed-phase HPLC analysis as described earlier [14]. Detection of absorbance was monitored at 214 nm only. An accurately weighed amount (100 µg) of pure emericellipsin A was placed in a 5 mL volumetric flask and dissolved in 50% water MeCN methanol (Panreac VWR, Spain, Hungary) to produce a 100-µg mL^−1^ standard stock solution. This stock solution was diluted serially with methanol 50% water MeCN, revealing a solution series with the concentrations of 33.751—235.025 µg mL^−1^ (injection volume of 40 µL). Three replicates were performed, and the mean of the recorded values was used for the calibration. 

The resulting calibration curve was used to express the EmiA production (Appendix A). 

#### 4.3.3. Sample Preparation and MALDI-TOF MS/MS Analysis 

For MALDI-TOF MS analysis, a 0.3 μL EtOH (50% in MQ)-solution or MeCN-H_2_O fraction (collected during HPLC separation) sample and 0.5 μL of 2,5-dihydroxybenzoic acid (Sigma-Aldrich, Germany) solution in 20% MeCN + 79.5% water (MQ) + 0.5% TFA (HPLC, Sigma-Aldrich, Germany) in a concentration of 20 mg/mL were mixed on a target of the spectrometer. The spectra recording and MS analysis was carried out with utilization of a MALDI-TOF MS spectrometer (UltrafleXtreme BrukerDaltonics, Bremen, Germany) equipped with a UV-laser (Nd) in the mode of positive ion registration with the utilization of reflection. The accuracy of mass detection was about 1 Da.

#### 4.3.4. EmiA and dEmiA Form Detection Assay 

EtOH (50% in MQ)-extract was separated with the HPLC method and as a parallel assay, the fraction related to the corresponding peaks on a chromatogram was collected in a test tube for further MALDI-TOF MS detection. EtOH (50% in MQ)-extracts were centrifuged with Beckman Coulter Microfuge^®^ 22R Centrifuge under the room temperature at the rate of 14.000 rpm for 7 min. Then, the pellet was eliminated, and the supernatant was utilized to perform HPLC analysis.

All structures were obtained using ChemDraw 12.0 (PerkinElmer) software and the MALDI-TOF MS and MS/MS spectra were processed with Daltonics flexAnalysis 3.4 (Bruker) software. Chromatograms were obtained with Milichrom A-02 (Econova) software utilization and processed with Multichrom for Windows 9x&NT version 1.5x-E” (Ampersend, Moscow, Russia) software.

### 4.4. Biological Assays

#### 4.4.1. Antifungal Activity

The antifungal activity of a crude peptaibol extract was measured by the disc-diffusion method. Discs with a 6 mm diameter containing 40 µL of the sample were deposited on PDA agar plates (Sigma-Aldrich, St. Louis, MO, USA). The diameter of the inhibition zones was measured after 24 h at 28 °C. Amphotericin B solution (Sigma-Aldrich) was used as a positive control. The minimal inhibitory concentration (MIC) value of each individual compound was determined using the broth two-fold microdilution method according to CLSI/NCCLS documents M27-A3, M38-A, and M38-A2 [54]. Yeast strains *Candida albicans* ATCC 14053 and the fungal strain *Aspergillus niger* ATCC 16404 were obtained from the American Type Culture Collection (ATCC, Manassas, VA, USA).

#### 4.4.2. Real-Time Cell Analyzer Test

Real-time cell line proliferation analysis was performed using the A colorectal carcinoma (HCT116) cell line (ATCC^®^ CCL-247TM) [15,48,49]. After thawing, the cells were passaged two or three times. The cells were then cultured in DMEM supplemented with 10% fetal bovine serum in a cell incubator at 5% CO_2_ and 37 °C in the iCELLigence RTCA system. Suspensions of HCT116 cells were prepared in DMEM medium, and the density was adjusted to 1 × 10^5^ cells/mL. Then, 300 µL of the cell suspension was added to the wells of the device. A day later, the old nutrient medium was replaced with a new medium containing various concentrations of EmiA and a medium without peptides for the control. The cells continued to be incubated for another two days. The CI value was set by the RTCA software package based on the impedance signal [16].

## Figures and Tables

**Figure 1 molecules-27-01736-f001:**
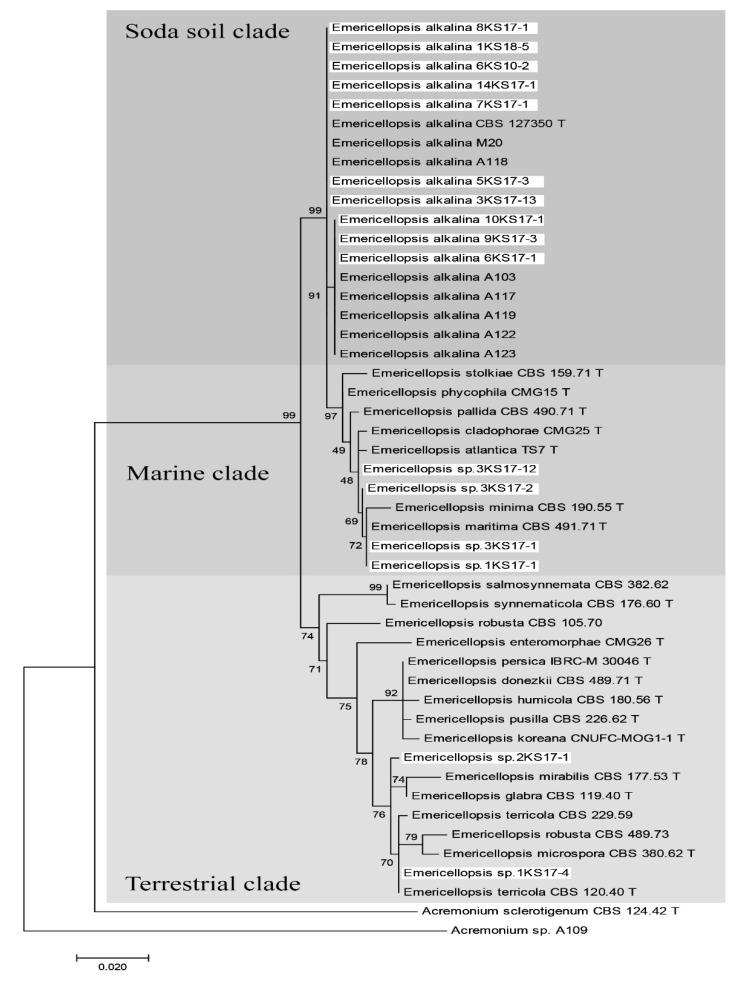
Maximum likelihood tree for the *Emericellopsis* genus based on partial sequences for the ITS rDNA (including 5.8S rDNA) region. Branch lengths are proportional to the estimated number of nucleotide substitutions. The BP values are displayed on the nodes (BP; 1000 replicates). *Emericellopsis* spp. and related species were clustered into a “Marine”, “Soda soil”, or “Terrestrial” clade. Taxa names of the isolates obtained in this study are highlighted in a light color. “T” beside each strain name indicates the strains as the ex-type strain.

**Figure 2 molecules-27-01736-f002:**
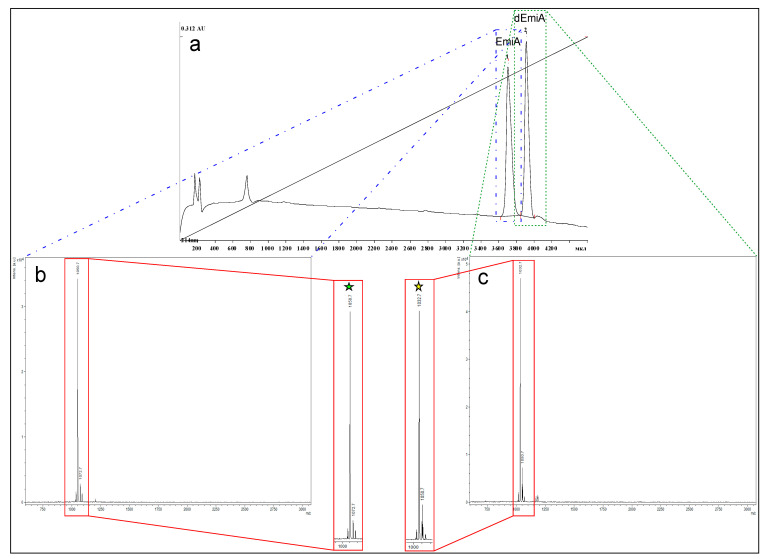
(**a**) Chromatogram of the analyzed sample with the HPLC method. Only two peaks, 1 and 2, were visualized with retention times of 37.4 min and 39.3 min, respectively (on the chromatogram values are presented in volume units). The names of probable compounds are marked at the top of the corresponding peaks. Both peaks are identified as dotted rectangles colored in blue and green and were related to the corresponding MALDI-TOF MS spectra. (**b**) The MALDI-TOF MS spectra of the chromatographic fraction related to the peak with a retention time of 37.4 min. (**c**) The MALDI-TOF MS spectra of chromatographic fraction related to the peak with a retention time of 39.3 min. In the red rectangles, the zoomed spectra of the masses are represented, and the major peaks related to the masses of 1050.7 Da and 1032.7 Da, which correlate to the masses of EmiA and dEmiA, which are marked with a green and with yellow star, respectively.

**Figure 3 molecules-27-01736-f003:**
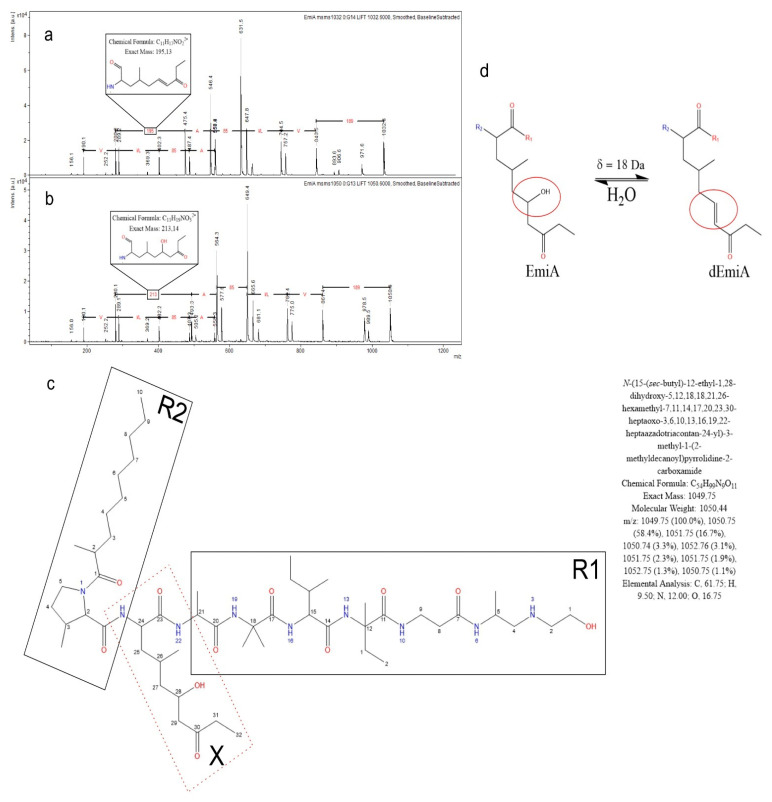
(**a**,**b**) The MALDI-TOF Ms/Ms spectra of the molecular ions with masses of 1032.7 Da and of 1050.7 Da are respectively depicted. Molecular mass differences of fragments is presented as a straight black line with the annotation of masses colored in red. The most unlikely fragment with the molecular formula and monoisotopic mass are highlighted in the black rectangle on both spectra. (**a**) Fragment relates to the 2-amino-6-hydroxy-4-methyl-8-oxodecanoic acid (AHMOA) residue. (**b**) Fragment relates to the 2-amino-4-methyl-8-oxodec-6-enoic acid (AHMOEA) residue (**c**). The structure of EmiA is presented without stereocenter indication. The residue of interest in the structure of EmiA was named as X and highlighted in the red dotted rectangle. The right and the left sets of the residues are highlighted in the black rectangles and named as R1 and R2 radical, respectively. Physico-chemical information about the full name of EmiA, its exact molecular weight with the condensed molecular formula, and information on the atom composition is depicted in the high right corner of (**d**).

**Figure 4 molecules-27-01736-f004:**
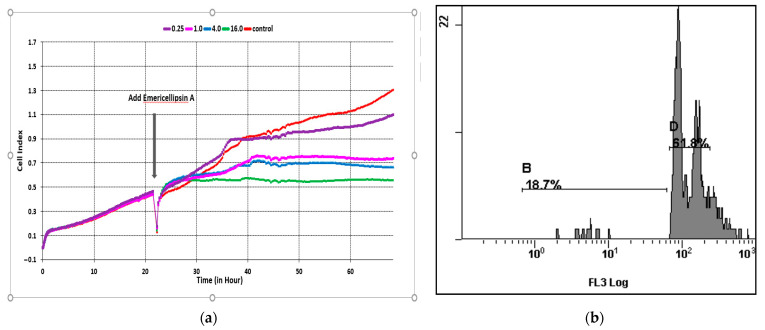
(**a**) Cell index of the HCT116 cell line at different concentrations of emericellipsin A: 0.25 μg/mL; 1.0 µg/mL; 4.0 µg/mL; 16.0 µg/mL. (**b**) The level of apoptosis of HCT116 cells under the influence of emericellipsin A at a concentration of 0.25 μg/mL: zone B—cells with fragmented DNA (apoptosis); zone D is the percentage of cells in different phases of the cell cycle.

**Table 1 molecules-27-01736-t001:** *Emericellopsis* strains from soils with different types of mineralization involved in the study. The total amount of emericellipsin A in crude extracts of culture broth and mycelium.

№	No. of Strains, No. of VKM, VKPM, CBS Collections	pH of the Soil	Total Salts (g/kg)	EmiA, CB, mg/L	EmiA, M, mg/L	Presence of Homologues B–E
**Soda soils**						
1	E101, F-4108, CBS 127350 T	10.1	73	429.5	173	d.
2	A112	10.1	33	26	18	n.d.
3	A113, FW-1476	11	57	358.75	356.5	d.
4	A120	9.9	310	28.5	162.5	n.d.
5	A121	10.2	73	140.25	84.75	n.d.
6	A123			219.5	157.25	d.
7	A124	10.1	60	166.75	266.25	n.d.
8	A125	10.1	7.1	71.25	303.5	d.
9	A126	10.1	1.9	158	85.5	n.d.
10	A127	10.1	1.9	154.75	160.5	d.
11	M14, F-3905, CBS 120043	9.9	310	28.5	26	n.d.
12	5KS17-3	11.0		30.5	74.25	n.d.
13	6KS17-1	11.0		140	268.5	d.
14	7KS17-1	11.0		15.5	42.25	n.d.
15	8KS17-1	11.0		104.5	283.5	d.
16	9KS17-3	11.0		0	66	n.d.
17	10KS17-1	11.0		136	56.25	n.d.
18	14KS17-1	11.0		33	128	d.
19	6KS10-2	9.8	48	27.75	103.5	d.
**Soda-chloride-sulfate soils**						
20	A103	9.6	100	28.5	321.75	d.
21	A116	9.6	100	25.75	338.75	d.
**Chloride soils**						
22	A114, FW-1473	10	187	85	116	n.d.
23	A122	9.5	65	101.25	129.5	n.d.
24	3KS17-13	8.0	350	68.5	136	n.d.
25	1KS17-1 *	8.5	200	20.25	295.5	n.d.
26	1KS17-4 *	8.5	200	n.d.	n.d.	n.d.
27	2KS17-1 *	7.5	350	32.75	85.25	n.d.
28	3KS17-1 *	8.0	350	n.d.	n.d.	n.d.
29	3KS17-2 *	8.0	350	n.d.	n.d.	n.d.
30	3KS17-12 *	8.0	350	n.d.	n.d.	n.d.
**Chloride-sulfate soils**						
31	A115, FW-1474	9.6	225	61	67.25	n.d.
32	A117, FW-1471	9.9	53	61	231.25	n.d.
33	A119	10.1	38	75.5	152.5	n.d.
**Soda-chloride soils**						
34	A118, VKPM F1428	9.6	137	262	184	d.
35	M20, FW-3040,CBS 120044	9.6	137	342	202	d.
**Sulfate-soda soils**						
36	A128	10.3	139.4	74.5	249.75	d.
37	M71, F-3907, CBS 120049	10.3	139	112.25	55	d.
**Undefined**						
38	1KS18-5	9	3	131.5	100.5	n.d.

All isolates are *E. alkalina*, except those marked with an asterisk (*).

## Data Availability

All sequence data are available in NCBI GenBank following the accession numbers in the manuscript.

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
