# Peer review of "Exploring Peptaibol’s Profile, Antifungal, and Antitumor Activity of Emericellipsin A of *Emericellopsis* Species from Soda and Saline Soils"

_molecules, 2022, doi:10.3390/molecules27051736_

Round 1
Reviewer 1 Report
This paper has identified EmiA (an antifungal peptaibol) production from many Emericellopsis strains, and found that the extracts could suppress the growth of fungal pathogens and HCT cells. The data is solid. As for the antifungal experiments, it would be better to add several typical plate pictures.
L443, 'was caddied out', modified to 'was carried out'.
Author Response
Dear Reviewer,
Thank you for your revision. The photo of antifungal activity is included in the Supplement of the manuscript. L 433 line is corrected.
Reviewer 2 Report
- Please add the unit of MIC.(Line 219)
- Did you measure the effect of dEmia on the growth of HCT116 cells?
Author Response
Dear Editor,
Thank you so much for your revision. All MIC data is included in the revision version of the manuscript. Unfortunately, in this manuscript, we did not test deEmiA on any cancer line, but we plan to do so in further experiments.
Reviewer 3 Report
Anastasia et al., described the manuscript entitled, “Exploring Peptaibol’s Profile, Antifungal and Antitumor Activity of Emericellipsin A of Emericellopsis Species from Soda and Saline Soils.” Authors described the manuscript well. In this manuscript authors isolated Emericellipsin A species and evaluated their antifungal as well antitumor activity. Interestingly it was inhibited cell growth and division by suppressing proliferation, causing cell cycle in the G0/1 and G2/M phases.
This manuscript can be accepted for publication after making grammatical corrections.
Author Response
Dear Reviewer,
Thank you so much for your revision! The grammatical mistakes of this manuscript have been corrected in some places.